# Reflective Functioning in Patients with Irritable Bowel Syndrome, Non-Affective Psychosis and Affective Disorders—Differences and Similarities

**DOI:** 10.3390/ijerph18052780

**Published:** 2021-03-09

**Authors:** Larisa Dzirlo, Felix Richter, Dagmar Steinmair, Henriette Löffler-Stastka

**Affiliations:** 1Department of Internal Medicine and Psychosomatics, Krankenhaus der Barmherzigen Schwestern, 1060 Vienna, Austria; larisa.dzirlo@chello.at; 2Department of Psychoanalysis and Psychotherapy, Medical University of Vienna, 1090 Vienna, Austria; Felix-D-Richter@gmx.de (F.R.); dagmar.steinmair@meduniwien.ac.at (D.S.); 3Karl Landsteiner University of Health Sciences, 3500 Krems, Austria

**Keywords:** irritable bowel syndrome, mentalizing, reflective functioning, psychosis, affective disorders

## Abstract

Irritable bowel syndrome (IBS), as part of the functional somatic syndromes, is frequent in the general population. Medical care and morbidity costs are high, and so is the psychological and somatic strain. The etiopathogenesis of IBS is still poorly understood; it is assumed to be multifactorial and to include biopsychosocial factors. Links between the intestine, psyche, nervous system (e.g., via the hypothalamic–pituitary–adrenal axis (HPA-Axis/neurotransmitters) and with the microbiome, the immune system have lately been investigated. Factors such as personality traits, mentalization, and early attachment strategies (deactivating and hyperactivating) have been suggested to influence IBS with relevance for treatment regimens. At this time, data on reflective functioning (RF) is lacking. Within a cross-sectional, we examined the mentalizing capacity of a clinical sample (*n* = 90) consisting of patients with IBS (*n* = 30), affective disorders (AD; *n* = 28), and non-affective psychosis (NAP; *n* = 32). The reflective functioning scale was used based on the brief reflective function interview (BRFI). The results revealed severe impairment in patients with IBS concerning their mentalizing ability, which was comparable to patients with affective disorders. Patients with non-affective-psychosis showed the lowest mentalizing ability. Thus, psychotherapeutic treatment with a focus on mentalization could be a promising approach.

## 1. Introduction

Mentalizing describes a particular facet of the human imagination: an individual’s awareness of mental states in themselves and the other people, particularly in explaining their actions, feelings, thoughts, beliefs, and wishes that demonstrate what people do [1]. There is a strong link between attachment, mentalization, and stress regulation. Secure attachment experiences play a quintessential role in developing the stress system and the development of resilience when faced with adversity [2]. Hypomentalizing is characteristic of the modus of “psychical equivalence”. The patients who are functioning in this mode equalize the internal and external world and use the principle of attachment deactivating strategy [3]. Narratives of this kind appear inflexible and simplistic [4]. As compared to hypomentalizing, hypermentalizing is understood as an over-attribution of intentions [5], which can be described as pseudo-mentalizing or pretend mode functioning [4]. Teleological thinking reflects the presence of unmentalized self-states, which get externalized and regulated interpersonally [4].

Since the 1990s, the concept of mentalizing has gained intense popularity in clinical research [4]. Adding significant knowledge to our understanding of borderline personality disorders (BPD), mentalizing is nowadays going beyond a distinct description of one specific disorder, moving to a transdiagnostic concept [6]. Reflective functioning was researched in various psychopathologies [7], including BPD [8], panic disorder [9], depression, psychosis [10], and eating disorders [11].

IBS is a widespread (prevalence of 11.2% [12]) functional gastrointestinal disorder. Thus, depending on the subtype of IBS, symptoms, such as diarrhea, constipation, pain, or bloating, in the abdomen occur unrelated to any organic gastrointestinal disorder. The economic costs of diagnostic procedures, the psychological and somatic strain of patients are very high. Patients often feel misunderstood and stigmatized and lack adequately developed coping strategies.

The etiology of IBS includes biological factors such as disorders of the hypothalamic–pituitary–adrenal axis (HPA-Axis), immune system, neurotransmitters, and microbiome, as well as the influence of psychological factors such as pathological personality traits, mentalization, and early attachment strategies (attachment-deactivating and attachment-hyperactivating). In IBS, either HPA-Axis hypoactivation or HPA-Axis hyperactivation can occur; more precisely, a switch of the HPA axis from a state of ‘overdrive’ to ‘underdrive’ is observed [13]. HPA-axis hyperactivation is characteristic for depression [14], anorexia nervosa [15], panic disorder, sexual abuse [16] and psychotic disorders [17,18]. HPA-axis hypoactivation is characteristic for post-traumatic stress disorder (PTSD) [19], chronic fatigue syndrome (CFS) [20], fibromyalgia [21], and probably diarrhea dominant IBS-D [22]. There is also high comorbidity with other psychiatric diseases such as depression [23] and anxiety [24]. It is not easy to conclude if psychiatric disorders are the cause or consequence of functional somatic diseases such as IBS.

The treatment of IBS is symptomatic, and there is no well-established therapy. The patients are often considered to be “difficult to treat”. In countertransference, the professionals often feel very helpless [25], bored and empty, and patients are not capable of verbalizing their feelings. The thinking mode of patients seems to be concrete and is less related to their own bodies. These observations confirm the Theory of Pierre Marty and Michel de M’Uzan from French Psychosomatics School about “operative thinking” (“pensée operatoire”) and “bland relations” (“relation blanche”) in psychosomatic patients [26]. It is known that the process of mentalization is linked with the process of somatization. There is data, for example, that show the mentalization impairment in patients with other functional somatic disorders (FSD), such as chronic pelvic pain syndrome [27], fibromyalgia [28], or somatoform disorders [29].

The study hypothesis is that the patients with IBS have altered mentalizing ability, with the specific mentalizing modi of IBS patients still to be defined and investigated. We hypothesize that IBS patients’ reflective functioning (RF) is low as RF in psychotic disorders and much lower than RF in depressive disorders.

## 2. Materials and Methods

The current study is part of the ongoing BiTMeP (Bindung, Trauma, Mentalisieren, Psychosen (Attachment, trauma, mentalizing, psychoses)) investigation, a multicentric longitudinal study conducted at the Klinikum am Wörthersee at the psychiatric department and the child and adolescent psychiatry Klagenfurt (AT) as well as at the psychiatric department at Kaiser-Franz-Josef Hospital, Vienna (AT). The project received ethics approval from the ethics review board at Klagenfurt (AT) and Vienna (AT). The IBS Patients were recruited at a Viennese internal-psychosomatic medical department (i.e., Krankenhaus der Barmherzigen Schwestern). A cross-sectional observational, descriptive study was performed to explore differences in reflective functioning between patients with non-affective psychosis, IBS, and affective disorders.

### Recruitment and Participants

IBS patients: A total of 30 patients already diagnosed with IBS according to the Rome IV criteria by experts were contacted via the health care center for IBS and voluntarily participated in the study.

The participants’ ages ranged from 21 to 60 years. Thirteen participants (43.3%) were males and 17 (56.6%) females (sample characteristics are shown in Table 1).

Inclusion criteria: all participants were psychotherapy naïve and did not take any psychopharmacological medication.

Exclusion criteria: lacked informed consent to participate in the study, age <20 or >60 years, a history of inflammatory bowel disease or history of mental disorder and psychotic episodes, severe substance addiction, or neurological limitations. Patients who were not fluent in the German language were also excluded (the language of the interview and applied test).

Non-affective psychosis group: eligible participants for the non-affective psychosis group were 32 adults aged from 16 to 70 with a clinical diagnosis of psychosis-spectrum-disorder including schizophrenia (ICD-10 [30] F20), schizoaffective (ICD-10 [30] F25) and acute transient psychotic disorder (ICD-10 [30] F23). All participants were diagnosed by a psychiatric clinician using ICD-10 criteria [30]. They were inpatients at psychiatric hospitals in Vienna and Klagenfurt (AT). Fourteen inpatients were female (43.8%), and 18 were male (56.3%).

Exclusion criteria: lacked informed consent, severe substance addiction, or neurological limitations. Patients who were not fluent in the German language were also excluded (the language of the interview and applied test).

Affective disorders group: patients for the affective disorders group were 28 adults aged over 20 to 74 with a diagnosis of unipolar/bipolar depression with or without comorbidity of anxiety disorders or a cluster C personality disorder. All participants were diagnosed by a psychiatric clinician using ICD-10 criteria [30]. They were inpatients at psychiatric hospitals in Vienna and Klagenfurt (AT). Seventeen outpatients were female (60.7%), and 11 were male (39.3%).

Exclusion criteria for this group were lack of informed consent, psychotic episodes, a diagnosis of IBS or inflammatory bowel disease, a history of mental disorder, severe substance addiction, or neurological limitations. Patients who were not fluent in the German language were also excluded (language of the interview and applied test).

All participants gave written informed consent; the participants received no payments.

## 3. Measures/Instruments

### 3.1. Clinical Interview

#### Brief Reflective Function Interview (BRFI)

The BRFI was published by Rudden, Milrod, and Target (2005) and is designed to assess reflective functioning [31]. It is a semi-structured interview consisting of 10 questions focusing on attachment-related contexts. The interviewer asks the proband to reflect about one parent of their choosing and about one person in their current life to prove the patient’s attachment capability in nonparental relationships.

The questions are divided into two types, those that permit the patient to demonstrate their reflective-self capacities, so-called “permit” questions, versus “demand” questions which demand the probands demonstrate their reflective-self ability [32]. The “permit” questions are coded with the value < 4 and >4 (examples English version). The interview takes about 15–30 min.

It was developed as an alternative to the adult attachment interview (Main, George, and Kaplan, 1985), which is, due to its complexity, hard to integrate into bigger sample sizes. The BRFI was shown to be a reliable and valid alternative to the AAI (Adult Attachment Interview). Whereas the AAI can be used to assess reflective functioning and attachment representations, the BRFI can only be used to assess RF due to its focus on reflecting attachment figures and leaving out biographical episodes. The interviews were recorded and transcribed afterward to be analyzed by the Reflective Functioning Scale [32].

### 3.2. Clinical Ratings (Based on the Interview)

#### 3.2.1. The Reflective Functioning Scale (RFS)

The Reflective Function Scale was developed by Peter Fonagy and colleagues (1998) to measure the capacity to mentalize thoughts, intentions, feelings, and beliefs about oneself and others [32]. The scale is numbered with 11 points, from −1 to 9. Marked bizarre, unintegrated, and inappropriate explanations are marked with −1, while exceptional and sophisticated explanations are scored with 9. 

#### 3.2.2. Rome IV Criteria

IBS, according to Rome IV, is defined as a recurrent abdominal pain on average of 1 day/week in the last three months associated with two or more following criteria:
Related to defecationRelated to the change in frequency of stoolAssociated with a difference in the form (appearance) of stool

According to symptoms, the patients were divided into three groups: patients with IBS with diarrhea, mixed, or constipation (IBS-D, IBS-M, IBS-C) [33].

### 3.3. Statistical Analysis

SPSS 21 was used for statistical analysis. The Shapiro–Wilk test was used to test for normally distributed residuals. It indicated non-normally distributed data; therefore, the nonparametric Kruskal–Wallis test and chi-square test were used for between and within-group comparison. Due to the non-normally distributed data, the Spearman correlation was used to test for correlations between age and RF. Cohen’s d was calculated for significant results. The significance level was *p* < 0.05 (two-tailed) for all analyses.

## 4. Results

### 4.1. Sample Characteristics

IBS patients in our sample were between 21 and 57 years old (mean age = (40.8), SD (11.3)). Seventeen (56.6%) were female, and 13 (43.3%) were male. In the non-affective psychosis (NAP) sample, the patients were from 16 to 70 years old (mean age = (38.2), SD (16.3). Fourteen (43.8%) were female, and 18 (56.3%) were male. The patients of the affective disorders (AD) sample were from 20 to 74 years old. Seventeen were female (60.7) and 11 were male (39.3) (sample characteristics are shown in Table 1).

There were no significant differences between groups regarding sex (χ^2^ = 1.92, *p* = 0.381) as well as regarding age (Kruskal–Wallis χ^2^ = 2.17, *p* = 0.339). There were no significant correlations between RF and age between groups.

### 4.2. Comparison between Groups Regarding RF

There was a statistically significant difference when confronting groups and RF (Kruskal–Wallis χ^2^ = 17.36, *p* < 0.001), with a mean rank of 32 for the NAP, 58 for the AD, and 52 for the IBS. Dunn–Bonferroni-Test was used as a post-hoc test. It shows the statistical significance of differences in groups regarding RF for NAP and AD (z = −3.964, *p* < 0.001) as well as NAP and IBS (z = −3.013, *p* = 0.008). There were no significant differences between IBS and AD regarding RF (z = 0.990, *p* = 0.322). Effect sizes according to Cohen (1992) for differences between NAP and AD were r = 0.512 and for NAP and IBS r = 0.382 [34]. Thus, an effect size of 0.512 corresponds to a strong effect, and 0.382 corresponds to a medium effect [34] (see Table 2).

### 4.3. Comparison between Sub-Groups Regarding RF

#### 4.3.1. IBS

RF-scores for IBS-sub-groups are shown in Figure 1. There was a statistically significant difference when confronting IBS sub-diagnosis and RF (Kruskal–Wallis χ^2^ = 8.41, *p* = 0.015), with a mean rank of 11.63 for IBS-M, 18.05 for IBS-D, and 24.63 for the IBS-C. The Dunn–Bonferroni-Test was used as a post-hoc test. It shows the statistical significance of differences in RF regarding IBS-M and IBS-C (z = −2.669, *p* = 0.023). There were no significant differences between IBS-M and IBS-D (z = −1.829, *p* = 0.67), as well as no differences between IBS-D and IBS-C regarding RF (z = −1.276, *p* = 0.202). The effect size for differences between IBS-M and IBS-D was r = 0.597. Thus, the effect size of 0.597 corresponds to a medium effect [34].

#### 4.3.2. NAP

Non-affective psychoses showed significantly different RF-scores (Kruskal–Wallis χ^2^ = 6.663, *p* = 0.036), with a mean rank of 14.55 for schizophrenia, 26.2 for acute transient psychotic disorders, and 15.25 for schizoaffective disorders. The Dunn–Bonferroni-Test was used as a post-hoc test.

It shows the statistical significance of RF differences regarding the diagnosis of schizophrenia and acute transient psychotic disorders (z = −2.554, *p* = 0.032). There were no significant differences between schizophrenia and schizoaffective disorders (z = −0.166, *p* = 1.000) and no significant differences between schizoaffective disorders and acute transient psychotic disorders regarding RF (z = 1.973, *p* = 0.146). The effect size, according to Cohen (1992), for differences between schizophrenia and acute transient psychotic disorders was r = 0.501. Thus, the effect size of 0.501 corresponds to a strong effect.

#### 4.3.3. AD

There was no statistically significant difference between RF in IBS and RF in affective disorders (Kruskal–Wallis χ^2^ = 1.632, *p* = 0.442).

## 5. Discussion/Conclusions

The current study’s goal was to assess the mentalizing capacity in patients with IBS in comparison with affective disorders and patients with non-affective psychosis. The hypothesis was that this capacity would be low compared to patients with affective disorders, significantly lower in RF, and similar in RF as NAP patients. Because there is hardly any research on IBS and mentalizing, the study had an exploratory character, trying to bring light in the appearance of RF in IBS. Furthermore, the within-group comparison aimed to show differences or similarities within disease entities.

To our knowledge, this study was the first one investigating reflective functioning with the BRFI and RF in IBS patients.

Agostini et al. (2019) investigated the association between attachment and mentalizing in patients with Inflammatory Bowel Disease (IBD) in comparison to healthy controls [35]. Although IBS and IBD are two different disease entities, they share similarities in symptomatic as well as affection by psychological factors [36]. The results indicated that IBD patients show higher attachment anxiety and lower scores on mentalization. Interestingly the IBD patients performed worse on the eyes test, but similar in the self-reported mentalization compared to healthy controls [35]. Smith et al. (2019) highlighted the importance of affective awareness regarding pain intensity and somatization in IBS patients [37]. According to this assumption, the affective dimension in mentalization could be a possible therapeutic factor in treating IBS.

In contrast to IBS, the mentalization deficits in psychotic disorders had been well explored and mainly operationalized by the cognitive-oriented theory of mind.

Debbané et al. (2016) are focusing on a specific aspect of the mentalizing deficits in psychotic patients, speaking of so-called embodied mentalizing [18]. Embodied mentalizing means a capacity to monitor one’s own bodily sensory-affective signals and critically reflect on them. Psychotic patients are known to show difficulties in self-oriented mentalizing, which was explored within the concept of source monitoring [38]. Auditory hallucinations, for instance, reflect on the inability to monitor one’s mental states and differentiate them from external stimuli (Keefe et al., 2002). The distorted evaluation of affective/cognitive states could, therefore, lead to the experience of feeling controlled by others [39].

Beneath the presence of the mentalizing deficits in psychotic disorders, its influence on the etiology of psychosis and the possibility as a therapeutic factor is examined [18].

Deficiencies had been shown in negative symptoms [40], positive symptoms [41] as well as social cognitive deficits [42,43]. Studies that include attachment-related mentalizing, using the RFS, are limited. MaBeth et al. (2011) researched a sample of 34 patients with a first-episode psychosis using the RFS [10]. Results showed a low to questionable RF (median = 3). These deficits are discussed within a state-trait debate, whether these deficits are caused by psychotic symptoms (state) or persist throughout the disease. Boldorini et al. (2020) tested the predictive value of RF for the development of psychosis in a sample of patients at ultra-high risk for psychosis (UHR) [44]. The UHR patients scored, on average, a median of 2.02, whereas the clinical control group scored 3.44. Furthermore, a low RF raised the likelihood of developing a psychosis, and an RF score of 1.25 distinguished between patients who did or did not develop psychosis.

Studies focusing on mentalizing of affective disorders are limited. Fischer-Kern et al. (2013) examined the RF of 46 female inpatients with major depression in comparison with healthy control [45]. The patients with a significant depression scored significantly lower on RF (median = 2.4) than the healthy controls (median = 4.1). The deficit in mentalizing was general and not limited to depression-specific topics. On the contrary, Taubner et al. (2011) found no significant differences between chronically depressed patients (median = 4.0) and healthy controls (median = 3.6) but topic-related differences in RF concerning loss for the depressed patients [46].

For bipolar disorders, little is known about the capacity to mentalize. Bodnar and Rybakowski (2017) examined the cognitive and affective mentalization of patients with bipolar I at manic and depressed phases [47]. Results indicate that patients with bipolar I show deficits in both cognitive and affective mentalization. In the depressive phase, these deficits are proven to be correlated with cognitive impairment.

The goal of the study is to investigate the ability to mentalize in a sample of IBS inpatients in comparison to non-affective psychotic inpatients and inpatients with affective disorders.

Our findings show deficient reflective functioning of patients with IBS with a mean RF of 2.7, which corresponds to low or questionable RF (normal RF: 4; [48]).

In contrast to our hypothesis, RF in patients with IBS was similar to AD and significantly higher than NAP. Interestingly within IBS patients, there was quite a range of RF (mean = IBS-M = 2.1, IBS-D = 3.1, IBS-C = 4.4; shown in Figure 1) with a significant difference between IBS-M and IBS-C. Due to the small sample size, statements about these differences are limited, but because of the known linkage between mentalizing patterns and attachment [4], our findings could indicate different abilities to cope with attachment-related stress. According to Smith et al. (2020), affective mentalizing impacts experiencing pain and somatization in IBS. Therefore, research focusing on differences in mentalizing profiles between different diagnoses could be rewarding in finding more suitable treatments for IBS subtypes [37].

Our findings underline the mentalizing deficit in patients with NAP with a mean RF of 1.4. The RF is lower than in the sample of MaBeth et al. (2011) [10], which seems plausible because most of the patients in our sample had a long disease duration. As expected, RF was lower than in the sample of UHR patients [40] and close to the cut-off for the development of psychosis (median = 1.25). Our findings suggest that patients with acute transient psychotic disorders score significantly better on RF than patients with schizophrenia. These findings reflect the results from literature that mentalizing worsens throughout its course of illness [46].

Regarding the AD patients, RF was higher than in the female sample of Fischer-Kern with a mean of 3.3 but lower than in chronically depressed outpatients [45]. A mean of 3.3 still refers to questionable RF. There were no significant differences between the diagnoses of the AD sample. Interestingly, there was no significant difference regarding RF between patients with depression and bipolar disorder. Most of the patients with bipolar disorder had a depressed episode while interviewed. Results indicated that patients with depression and bipolar disorder do not differ in RF throughout depressed phases.

There is good evidence that patients with functional somatic disorder are characterized by high levels of early adversity and insecure attachment [49]. Individuals with insecure attachment rely upon so-called secondary attachment strategies in response to stress (i.e., attachment deactivating or hyperactivating attachment). The patients who use attachment deactivating strategies deny professionals’ help, their attachment needs, and they have little trust. They try to have autonomy and to pretend to be strong [48]. Attachment deactivating strategies are associated with HPA-Axis hypoactivity and disturbed immune functioning [50]. E. A. Mayer argues that individuals with IBS without constipation have a higher stress overload with hypoactivity of HPA-Axis [22]. The results of our study reveal that the mentalizing capacity is significantly lower in patients without constipation, which confirms the arguments of E.A. Mayer. Our analysis did not allow a statement whether patients with IBS-C have a higher reflective functioning; further research should be designed with a big enough sample size for specific subgroup analysis.

Our findings confirmed and replicated earlier studies and add new information about mentalizing in IBS patients (see [36]).

Further research should focus on the specific dimensions of RF in various disorders. Although assessing the different aspects of RF is still a difficult task, particularly by questionnaires to enable studies with broader samples [51], it is of great importance to clarify these replicated and novel deficits in RF to foster psychotherapeutic interventions.

If the mentalization deficit is a consequence or cause of IBS, it is still not clear. Further investigations are needed.

## 6. Limitations

Our results offer an overview of three distinct mental disorders regarding their ability to mentalize. Our study first examined mentalizing in IBS patients using the gold standard for assessing the capacity to mentalize (the RFS), which, therefore, provides strong validity. Beneath the novel findings concerning IBS, our study replicated and, therefore, strengthened the findings about mentalizing deficits in NAP and AD patients from earlier studies. Further, we put mentalization of IBS patients in reference to other mental disorders, which could be a starting point for further studies. Concerning the broad diagnostic entities (NAP, AD), the total sample size (*N* = 90) is relatively small, in particular, for the within-group comparisons. Due to the cross-sectional design, our findings should be cautiously interpreted for generalization.

However, we did not assess symptom severity in our study sample; further research should look at this risk factor for more severe psychological distress, as symptom activity in IBS is associated with psychological distress and mentalizing deficits [36].

## Figures and Tables

**Figure 1 ijerph-18-02780-f001:**
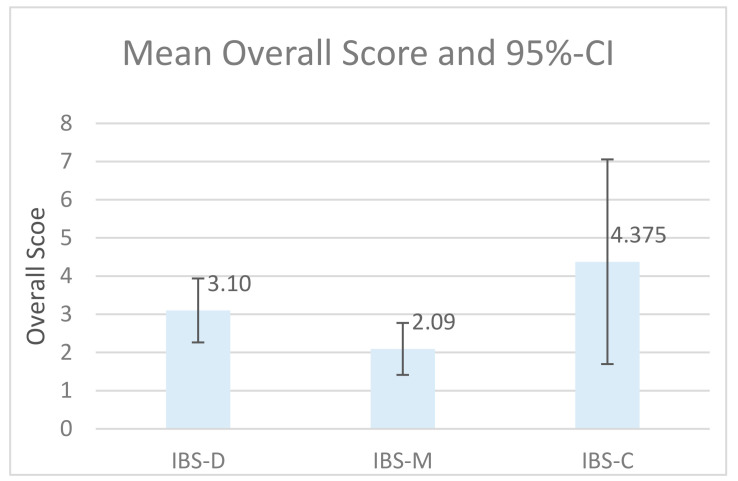
Reflective functioning (RF)-score for irritable bowel syndrome (IBS)-sub-groups.

**Table 1 ijerph-18-02780-t001:** Demographics and clinical data of the sample (*N* = 90).

Groups	NAP (*n* = 32)	IBS (*n* = 30)	AD (*n* = 28)	Statistics	*p*
Demographics					
Gender, *n* (%)				χ^2^ = 1.928	0.381
Male	18 (56.3)	13 (43.3)	11 (39.3)		
Female	14 (43.8)	17 (56.7)	17 (60.7)		
Age, mean (SD; range)	38.2 (16.3; 16–70)	40.8 (11.3; 21–57)	37 (13.8; 20–74)	χ^2^ = 2.166	0.339
**Diagnosis, *n* (%)**					
F20: Schizophrenia	21 (65.6)	-	-		
F23: Acute Transient Psychotic Disorder	5 (15.6)	-	-		
F25: Schizoaffective Disorder	6 (18.8)	-	-		
IBS-M	-	16 (53.3)	-		
IBS-D	-	10 (33.3)	-		
IBS-C	-	4 (13.3)	-		
F31: Bipolar Affective Disorder	-	-	12 (42.9)		
F32: Depressive Episode	-	-	4 (14.3)		
F33: Recurrent Depressive Disorder	-	-	12 (42.9)		
**Comorbidities, *n* (%)**					
PTSD	-	-	1 (3.6)		
Dependent PD	-	-	2 (7.1)		
Anorexia Nervosa	-	-	1 (3.6)		

Note. NAP: non-affective psychosis; AD: affective disorder; IBS: irritable bowel syndrome; IBS with diarrhea, mixed or constipation (IBS-D, IBS-M, IBS-C).

**Table 2 ijerph-18-02780-t002:** Reflective functioning of the groups (*N* = 90).

	NAP	IBS	AD	Cohen’s d	Statistics	*p*
	*n* = 32	*n* = 30	*n* = 28			
Mean (SD)						
RF by diagnosis						
	1.4 (1.9)	2.7 (1.4)	3.3 (1.7)	0.512 0.382	z = −3.964 z = −3.013	<0.001 0.008
specific study sub-groups						
F20	1.0 (1.6)			0.501	z = −2.554	0.032
F23	3.6 (1.9)					
F25	0.8 (1.5)					
IBS-M		2.1 (1.2)		0.597	z = −2.669	0.023
IBS-C		2.9 (1.6)				
IBS-D		2.7 (1.2)				
F31			3.5 (1.7)			
F32			2.3 (2.1)			
F33			3.4 (1.7)			

Note. NAP: non-affective psychoses; AD: affective disorder; IBS: irritable bowel syndrome; IBS with diarrhea, mixed or constipation (IBS-D, IBS-M, IBS-C).

## Data Availability

The data presented in this study are available on request from the corresponding author. The data are not publicly available due to privacy.

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
