# Peer review of "Reflective Functioning in Patients with Irritable Bowel Syndrome, Non-Affective Psychosis and Affective Disorders—Differences and Similarities"

_ijerph, 2021, doi:10.3390/ijerph18052780_

Round 1

Reviewer 1 Report

A very interesting article exploring the potential psychosomatic background of irritable bowel syndrome (IBS). The relationship between reflective functions, metallization and the occurrence of disorders such as IBS is poorly understood, therefore such works are very valuable. The study was conducted in a reliable and correctly described manner. The only drawback of these studies is the small and age-diversified study group, in future studies I would recommend that the study group be more homogeneous (including the ratio of men to women) and larger.

Author Response

Dear reviewer, thank you very much for your endorsement and the suggestions for further research, this is the line we try to follow, many thanks! kind regards, Henriette Löffler-Stastka

Reviewer 2 Report

The authors studied similarities of reflective functioning in patients with IBS, non-affective psychosis and affective disorders. The whole conception of the experiment seems interesting and reasonable. However, the article requires some changes before publication in the journal.

The some suggestions:
Abstract: The abstract should be a total max. 200 words. This abstract has more than 250 words and is with headings which are not required. Please, delete them. The authors revealed only IBS group results in the results without comparison between examined factors across all participants.
Lines 62-67 to my mind is unnecessary – the IBS criteria are well known.
Line 71 – please, verify hyper activating phrase – it should be written with a hyphen
Line 92-139 The authors raise the issue of mentalizing in some specific disorders. To my mind, the better will be adding this to the discussion section
Line 148 BiTMeP - it is some abbreviation? Please, add full name
Line 173: please, add the rference to ICD-10 criteria
In non-affective psychosis and affective disorders groups – please add the range of age
Line 182: Exclusion criteria for this group regarding affective disorder group or both affective and non-affective psychosis. If this regarding only to last group there were no exclusion criteria for the second group, if for second and thirdh, please add s to „group.”
In the healthy control group, the psychiatric diagnosis was not exclusion criterium?
Brief Reflective Function Interview (BRFI) [8] in the headline authors should not add the reference
Line 199 : AAI- please add information (full name) of scale
Line 221-238: Rome criteria – to my mind this information is not required. Only manner of patients division to IBS subtype could be describe. The authors did not include ICD-10 criteria for other diagnoses, so, RIV also is unnecessary.

Statistical analysis: The significance level was p < .05 (two-tailed) for all analyses
Why have authors not applied a correction for multiple comparisons, in case of analysis between three groups? In the methods authors describe results after Bonferroni correction – please add this information to the statistics section
Table 1 – the heading should be outside the table
p-value – please, verify the values
Table 1 and 2 are hard to read, there was no logical sense in diagnosis information – the information should be addedd according to divided diagnosis to specific study groups
line 259, 261 – p-value should not be 0 (p = .00), p could be lower than some number f.e. p<0.001
line 269 – to my mind sub-types is sounding better term
Table 2 is invisible, the p-value is probably incorrect values and in these tables, authors to my mind should also including results according to subtypes of disorders – specific study subgroups
Why did authors not examine f.e treatment effect, age or other sociodemographic/clinical data which could affect results? IT is necessary
Line 310-311 Interestingly within IBS patients there had been quite a 310 range of RF (mean= IBS-M= 2.1, IBS-D= 3.1, IBS-C=4.4; shown in Fig. 1) with a significant 311 difference between IBS-M and IBS-C. how to explain this result
Line 392 – Mean should be written in lowercase, sign. - please not use abbreviation
Limitations – in this section authors should also add the advanteges of the research
The section's length should be changes – the discussion should be more extended, and the introduction should consist of a smaller amount of word.

Author Response

Dear reviewer, many thanks for your helpful suggestions, we incorporated them accordingly, kind regards, Henriette Löffler-Stastka

Reviewer 3 Report

Thank you for this interesting secondary analysis that compares mentalizing, or reflective functioning (RF), in patients with irritable bowel syndrome (IBS, N=30), non-affective psychosis (N=32), and affective disorders (N=28), respectively. In this cross-sectional study, RF is assessed using the Brief Reflective Function Interview (BRFI), a 10-item semi-structured interview.

The study hypothesis, as stated by the Authors, has never been investigated before, and the Authors present original findings, which can add to the existing literature. The comparison of RF in IBS versus RF in non-affective psychosis is methodologically sound, in that mentalization deficits in psychotic disorders had been thoroughly investigated and constitute a valid term of comparison. The study of RF in affective disorders is still in ongoing.

My main concerns regard the background and the discussion of the findings:

As for the background, the Authors provide a clear and well-written description of mentalizing, yet they also describe at great lengths the role of the HPA axis in IBS (Introduction section, 4th paragraph). I would suggest synthetizing this paragraph, as the study provides no measure of stress and there is no reason to go into too much detail. The Authors could briefly mention that in IBS a hyperactivity or hypoactivity of the HPA axis can be observed as a result of a “switch” of the HPA axis system from a state of “overdrive” to “underdrive” (see: McEwen BS. Physiology and neurobiology of stress and adaptation: central role of the brain. Physiol Rev. 2007 Jul;87(3):873-904. doi: 10.1152/physrev.00041.2006. PMID: 17615391.; Chang L, Sundaresh S, Elliott J, Anton PA, Baldi P, Licudine A, Mayer M, Vuong T, Hirano M, Naliboff BD, Ameen VZ, Mayer EA. Dysregulation of the hypothalamic-pituitary-adrenal (HPA) axis in irritable bowel syndrome. Neurogastroenterol Motil. 2009 Feb;21(2):149-59. doi: 10.1111/j.1365-2982.2008.01171.x. Epub 2008 Aug 5. PMID: 18684212; PMCID: PMC2745840.)

As for the discussion of the findings, due to exploratory nature and the limitations of this study (cross-sectional design, small sample size, lack of assessment of potential confounders, inability to extend these findings to outpatients) the Authors should be cautious when suggesting the need of a psychotherapeutic treatment with a strong focus on mentalization. This, also in view of the fact that the effectiveness of psychotherapy in IBS is still unclear (and many studies show small to moderate effect sizes).

A minor comment regarding the Introduction section: please remove (or rephrase) the definition of IBS, as it’s repeated using the same words in the Methods section.

As regards the Methods section, the Authors state that the study has a “quasi-experimental design”, yet such design, per definition, “aims to establish a cause-and-effect relationship between an independent and dependent variable”. Also, when using this kind of design, researchers try to “account for any confounding variables by controlling for them in their analysis or by choosing groups that are as similar as possible”. This does not apply to the present study, one of reasons being that, besides gender and age, possible confounders (e.g., attachment, resilience, stress) were not assessed. I suggest removing this statement. Also, you state that you investigated RF in “inpatients” with IBS. Where these patients hospitalized due to this disorder? Please specify. You mention that 30 healthy controls were recruited as part of the study, yet this subsample is not mentioned elsewhere. If you don’t include it in the statistical analyses, consider removing it. Last, you mention twice (in the Clinical Interview and in the Clinical Ratings sections) that the BRFI was developed as a shorter alternative to the AAI. Please correct.

Please revise your paper with the help of a native English language speaker, due to:

  1. the use of colloquial language, e.g.: “the professionals often feel…bored and empty”, “The patients…try to have autonomy and to pretend to be strong”.
  2. the presence of many mistakes and typos, e.g.: replace “is wide spread” with “is a widespread”, spell out “there was no sign. difference”, replace “attachment hyper activating” with “attachment hyperactivating”.

Author Response

Dear reviewer, thank you for your helpful hints, we incorporated them accordingly, kind regards, Henriette Löffler-Stastka

Round 2

Reviewer 2 Report

The authors did not implement all of the suggestions. The manuscript should be improved according to the earlier proposal. Please, write a clearer answer to my review - it is hard to understand.

Author Response

Dear Reviewer, thank you for the suggestions, which we altogether included into the mansucript, we hope to have created an interesting paper for the readers of the journal, kind regards, Henriette Löffler-Stastka
